# Beyond individual responsibility: Exploring lay understandings of the contribution of environments on personal trajectories of obesity

**Nestor Serrano-Fuentes**[1]*, **Anne Rogers**[2], **Mari Carmen Portillo**[1]

**1** NIHR ARC Wessex, School of Health Sciences, University of Southampton, Southampton, United Kingdom, **2** School of Health Sciences, University of Southampton, Southampton, United Kingdom

* Nestor.Serrano-Fuentes@soton.ac.uk

**Data Availability Statement:** All relevant data are within the manuscript and its Supporting Information files.

## Abstract

### Introduction

Reversing the upward trajectory of obesity requires responding by including the multiple influences on weight control. Research has focused on individual behaviours, overlooking the environments where individuals spend their lives and shape lifestyles. Thus, there is a need for lay understandings of the impact of environments as a cause and solution to obesity. This research aimed to understand the influence of environments on the adoption of health practices in adults with obesity and to identify lay strategies with which to address environmental barriers to behaviour change.

### Methods

Nineteen adults with a history of obesity living in the United Kingdom were interviewed through video conferencing between May 2020 and March 2021. Semi-structured interviews and socio-demographic questionnaires were used, and data analysed through hermeneutic phenomenology informed reflexive thematic analysis.

### Results

Three main themes were created: living with convenience and normalcy: the increased accessibility of unhealthy food, people interacting with digital media for positive practice change, and the need to prioritise prevention in schools, the National Health Service and the food industry.

### Conclusions

The food environment was the major barrier, while interactions with social media was the most important opportunity to adopt healthy practices. The National Health Service was considered an obesogenic environment, something relevant since it has been traditionally recognised as an obesity management system. The perceptions from individuals with a history of obesity provide new suggestions on the influence of previously overlooked environments

**Funding:** The author(s) received no specific funding for this work.

**Competing interests:** The authors have declared that no competing interests exist.

to design more adequate and effective interventions and policies that consider, more than in the past, the environments where people spend their lives.

## 1. Introduction

The worldwide prevalence of obesity almost tripled between 1975 and 2016 [1], with an associated rise in other long-term conditions, including hypertension, coronary heart disease, joint and muscular disorders and type 2 diabetes (e.g. there is seven times greater risk of diabetes type 2 in people with obesity compared to those of healthy weight [2, 3]. Also, between 20 and 60% of people living with obesity suffer from mental health problems such as anxiety, clinical depression, and low self-esteem [4–6], which are consequences, among others, of the stigma and discrimination associated with obesity [7]. The rise in obesity levels is projected to continue estimating that by 2050, half of the adults in different countries (e.g. England) might have obesity [8]. In addition, this rise has led to an increase in health services expenditure. For example, the Organisation of Economic Co-operation and Development (OECD) predicts that the combined cost of obesity-related conditions in reducing life expectancy, National Health Service funds, and lost workforce productivity is £74 billion yearly [9].

It has been suggested that prevention efforts should focus on a multiplicity of factors [10]. However, most research on obesity has tended to focus exclusively on addressing individual behaviours and overlooking, for example, the environmental determinants of health outcomes in which individuals spend their lives and shape lifestyles and choices [11]. In the obesity literature, ecological approaches have synthesised findings which implicated economic, material/physical, social and political environmental factors with the physiological processes of excess fat [10, 12–15]. These approaches have addressed environments in different ways, for example, by researching food and obesogenic environments. The food environment refers to the settings in which a variety of food is accessible and available to individuals out-of-home in everyday life [16]. It influences how individuals buy and consume food according to aspects such as affordability, accessibility and availability of food, advertising, and media [17].

The obesogenic environment, which encompasses the food environment, refers to "the sum of influences that the opportunities, surroundings, or conditions of life have on promoting obesity in populations and individuals" [18]. It is divided into four broad categories of influence (economic, political, physical, and socio-cultural) [18, 19]. The economic environment refers to the costs related to food and physical activity (e.g. manufacturing or paid parking policies). The political environment describes regulations, laws and policies that influence obesity (e.g. the use of nutrition labels on packaged foods [20]. The physical environment includes 'what is available' in an environment (e.g. vending machines or cycle paths). The socio-cultural environment refers to a society's or community's beliefs, attitudes and values related to eating and physical activity (e.g. food and alcohol practices with friends to socialise). Furthermore, obesogenic environments are classified according to scale and dimension and are divided into micro-environments and macro-environments [18, 21]. Micro-environments are local (e.g. individual's home and workplace, recreational facilities, retail outlets, schools or sports clubs) and can be influenced by macro-environments or broader social structures (e.g. healthcare system, transportation sector, food industry, norms or mass media) [18, 22], which, in turn, affects individuals' health practices and behaviours. For example, the increase in petrol costs for transport (economic and macro-environment) might encourage the use of bike share schemes in cities (physical and micro-environment) and improve the levels of physical activity in individuals.

Most studies looking at mechanisms by which the environment influences behavioural physical activity, eating practices, and weight gain, have used cross-sectional study designs to answer whether and to what extent environmental characteristics are associated with these outcomes [23]. Research that has been done on environmental causes of obesity to date from a qualitative approach originates mainly from countries outside the UK and has focused exclusively on the food environment or the physical environment [17, 24–26]. Research in the UK has explored how children perceive their food and physical environments [27], low-income populations about the food environment [28, 29], or adults and the availability of urban parks and physical activity [30]. In addition, previous research shows the relevance of exploring lay views of the causes and management of obesity and identifying the environment's potential impact [31, 32]. For example, one study showed that while healthcare professionals identify structural and social factors as causes and solutions to obesity, the lay population seem to lean towards greater endorsement of biological and behavioural factors for causes and solutions [32].

In line with this, the present study aims to cover a research gap that asks to explore lay beliefs used by individuals [33, 34] to explain the wider notion of the obesogenic environment as a reason for ill-health in their everyday lives and enhance understanding of the interactions between these environments and individuals in the adoption and enactment of eating and physical activity practices. This will extend the focus to a more in-depth analysis of obesity-related practices in obesogenic environments independently of engagement with professional perspectives of specific causes and management strategies. Also, to our knowledge, examining the wider notion of the obesogenic environment in adults in the UK by exploring individual experiences has not been done before. Therefore, the aim of this research is to understand the influence of environments on the adoption of health practices in adults with obesity and to identify lay strategies with which to address environmental barriers to behaviour change.

## 2. Materials and methods

### 2.1 Design

This study used a hermeneutic phenomenology [35]. Phenomenology is the reflective study of lived experience about a phenomenon (in this case, the experience of living with obesity) [35], involving 'what' individuals experienced and 'how' they experienced it [36]. It is 'hermeneutical' because it is interpretive (rather than purely descriptive, as in transcendental phenomenology) [37]. The researchers' assumptions were not bracketed or set aside but were embedded and essential to the interpretive process [38] (an example can be seen in S1 Appendix, phase 5). This qualitative approach was used to identify environmental factors that may have been overlooked in the construction of previous theoretical ideas [39] and offer an in-depth understanding of the processes that underpin the interactions between individuals and the multiple environments in which obesity-related health practices occur [40].

### 2.2 Participants

A purposeful sampling technique [41] that involved snowballing was used because the aim was to find people who could address the research aim by providing a virtue of knowledge or experience, in this case, citizens who had the personal experience of living with obesity. Participants were considered eligible for inclusion in the study if they had a current or a history of body mass index (BMI) $\geq$30 kg/m2, lived in the UK, were able to communicate and understand English, had a device with Internet connection and microphone and an email address. They were approached in community settings and via social media (Twitter, Facebook and LinkedIn) using recruitments posters, online advertisements, sending letters of invitation to local weight management groups and long-term conditions related charities (asking for their

managers' approval when required) from England, Wales, Scotland, and Northern Ireland. Also, advocacy individuals from charities dedicated to empowering people living with obesity were contacted through the contact details of charities' webpages and Twitter and LinkedIn social media platforms.

## 2.3 Data collection

Individual semi-structured interviews [42] were conducted between May 2020 and March 2021 by NSF. An interview schedule guided the interviews with open-ended questions where language mattered [43]. Thus, the tone and the type of questions were reviewed (before starting the interviews) by the qualitative research group members of the authors' affiliation and pilot-tested with the two first participants (participants 1 and 2), who provided feedback in this regard after they conducted the interview. No changes were required in the interview schedule. The interviews lasted between 30 and 120 minutes (mean of 60 minutes) and were digitally recorded using a Dictaphone (Olympus WS-853). Furthermore, a socio-demographic questionnaire was used to register different participants' attributes and build relationships with the individuals' conceptions of health opinions and experiences.

Data was collected online due to the COVID-19 pandemic. Emails were used to provide information about the study to the participants. The interviewees sent confidential information using software (SafeSend) hosted by the research institution to transfer data across the network securely encrypted and ensure the utmost safety. Video conferencing was used to conduct the interviews since this method provides a more personable approach than others [44], which was pertinent to addressing this sensitive topic. Phone calls were considered at the participants' request or any technological problem. All interviews were held individually; eighteen were conducted by videoconference and one by phone call.

To mitigate potential biases, the researchers did not have any therapeutic relationship with the participants. Also, the researcher conducting the interviews (NSF) adopted a neutral role, focusing on asking the questions without providing personal opinions or data from the literature. This approach was employed to ensure that the participants' responses remained at the forefront and to encourage them to share their experiences and perspectives openly. To further promote transparency and create a safe space for sharing, the participants were informed about the researcher's neutral role and the importance of their honest responses through the participant information sheet provided before the study. This information was also reiterated just before commencing the interview on the day of data collection.

## 2.4 Data analysis

The interviews were transcribed verbatim, ten of them by the first author and the rest by a professional transcriber. The transcripts were uploaded to NVivo (version 1.2) to support the analysis. Reflexive thematic analysis [45] was taken using and adapting to this research a six-stage framework [46] to identify, analyse and report shared patterns of meaning across data. Data analysis followed an iterative and inductive process since the new themes and codes were created through the research, with movement back and forth between the different phases. The detailed analysis process with the corresponding phases can be seen in S1 Appendix.

Credibility, transferability, dependability, confirmability [47] and data adequacy [48] were the criteria introduced to establish the quality and rigour of data. Further information about how these criteria were met is explained in the section 'Quality and Rigour of Data' in the S1 and S2 Appendices, which shows two specific tools: one for evaluating thematic analysis manuscripts [46] and the consolidated criteria for reporting qualitative research (COREQ) [49]. The data richness and its in-depth and detailed analysis, together with data saturation,

determined the end of the analysis. After the seventeenth interview, no new codes and themes were generated from the narratives. Thus, it was concluded that the data analysis had reached a saturation point (data saturation) [50]. However, two more interested participants were interviewed to ensure and confirm that there were no new emerging codes and themes.

## 2.5 Ethical considerations

Ethical approval to conduct the research was obtained from the University of Southampton Ethics Committee under the reference number ERGO 55638. All the research conforms to the ethical principles for medical research on human beings set out in the Declaration of Helsinki [51]. Information provision (through a participant information sheet), informed consent, confidentiality and safety were fulfilled to promote the ethical and moral rights of autonomy, beneficence, non-maleficence and justice. Participants gave online written consent (due to the COVID-19 lockdown) before and verbally on the day of the interview with the option to withdraw at any point without any explanation. Consent was also provided by participants to audio record the interviews. Different practices were conducted to ensure the utmost confidentiality of individuals' data. For example, derived data were used on the questionnaire. This means that values/categories of a less granular nature were used to hide the exact values (e.g. age). All the real names were coded into fictitious names. Thus, research findings do not include information that can be used to directly identify any participant.

The Data Protection policy of the University of Southampton [52] was followed to protect personal data stored on computers or in an organised paper filing system. Electronic data collected (transcriptions of the anonymised interviews, questionnaires and signed consent forms) will be retained for the next ten years and will be kept on a University of Southampton password-protected computer, and audio files were already destroyed.

## 3. Results

Nineteen adults (thirteen women and six men) who have or used to have obesity were interviewed. Table 1 summarises the main participants' demographic factors.

Three main themes were created: *Living with convenience and normalcy: the increased accessibility of unhealthy food*, *people interacting with digital media for positive practice change*, and *the need to prioritise prevention in schools, the National Health Service and the food industry*.

## 3.1 Living with convenience and normalcy: The increased accessibility of unhealthy food

Participants identified the food environment as the main barrier to adopting healthier lifestyles, above all, through increased access and exposure to convenient food, triggering participants to eat unhealthier. For example, participant 14 explained the disproportionate amount and reduced price of processed products compared to healthy and fresh food. Participant 2 also pointed out the small number of healthy restaurants available compared to fast-food restaurants:

"Accessibility in supermarkets. There are two whole aisles with chips in a supermarket, another one only with cookies and chocolate. It's like oh my god, and ridiculous prices. It is cheaper to buy four doughnuts than buying lettuce". (Participant 14)

"There is far too much, far too much fast food and convenience food around. It's heart breaking that when you are going along the shops all you are seeing is things made of pastry and sugar and burgers and things like that. It's not very often you find a salad bar for

**Table 1. Demographic factors.**

| Demographic factors | | n (%) |
|---|---|---|
| Gender | Female | 13 (68%) |
| | Male | 6 (32%) |
| Age | 20–29 | 6 (32%) |
| | 30–39 | 4 (21%) |
| | 40–49 | 5 (26%) |
| | 50–59 | 3 (16%) |
| | 60–69 | 1 (5%) |
| Current BMI | 18.5–24.9 | 1 (5%) |
| | 25–29.9 | 5 (26%) |
| | 30–39.9 | 11 (58%) |
| | >40 | 2 (11%) |
| Place of residence | Deprived urban | 4 (21%) |
| | Affluent urban | 11 (58%) |
| | Affluent rural | 4 (21%) |
| Occupation | Nurse | 6 (32%) |
| | Healthcare assistant | 1 (5%) |
| | Health researcher | 2 (11%) |
| | Military services | 1 (5%) |
| | Physiotherapist | 1 (5%) |
| | Social worker | 1 (5%) |
| | Nursery supervisor | 1 (5%) |
| | Supplies manager | 1 (5%) |
| | Video editor | 1 (5%) |
| | Manager business | 2 (11%) |
| | Goods operator | 1 (5%) |
| | Student | 1 (5%) |
| Work situation | Unemployed | 3 (16%) |
| | Part-time | 5 (26%) |
| | Full-time | 10 (53%) |
| | Pensioner | 1 (5%) |

instance. It's much easier for our society to jump into quick food and it's unhealthy food". (Participant 2)

A discrepancy was found in the narratives when justifying shopping practices using a limited budget. Some of the participants preferred the purchase of unhealthy food since it is normally cheaper, full of high calories, and lasts longer than fresh and healthier food. For example, participant 4 described why she preferred to buy a big bag of processed chicken nuggets rather than vegetables and fruits. In contrast, participant 11, who is currently in healthy weight status, assured that eating healthy food is not more expensive. However, it is about adjusting personal practices and overcoming the structural incentives to buy less good food based on triggers from the environment:

"For example each month I will buy a big bag of chicken nuggets which has got 50/60 chicken nuggets in there because I know that's going to last me the month and it's going to do my kids a meal at least once a week. That's like £3 a bag. I can't get loads of fruit and veg for £3, or I can't make another meal for £3. It's weighing them up". (Participant 4)

"It's not expensive. It's about appropriating and being more knowledgeable what are your options really. . . so if I buy for instance beef one good example probably going to cost me £5 per kilogram and then if you buy chicken wings it's probably £2 for a kilogram. I would rather have the £5 per kilo beef that I know because it's a kilogram that will leave me with probably between 6 to 8 portions and that's going to sustain me for what four days for a fiver. And then you go to KFC (Kentucky Fried Chicken) or value meal of £6. It doesn't make sense. So I buy quality food and I only buy what I need to eat and anything I don't need to eat I don't buy them anymore". (Participant 11)

The demands of working life and employment inequalities were identified as causes of undermining lifestyles, involving greater burden and more effort on top of demands to earn enough to eat healthier. For example, participant 4 stated the difficulty of integrating healthy eating within her family (shift workers) due to the lack of time to cook; they just warmed pre-cooked food. Also, participant 5 recognised accessing unhealthy food after stressed and long days working at the hospital and a lack of resources of time and energy.

"Being a working family, I do my shift work and my husband works really weird shifts so he does 2pm until 11pm at night so he's not here when the kids get home, he's not here for tea-time. Living with my parents they do a lot of childcare so it's a kind of anyone and everyone does everything. My mum and dad both work so it's chuck something in the oven and for-get about it, leave it in for half an hour and then you've got dinner rather than prepare something". (Participant 4)

"Even if I have a day when I spend just collecting data we have got loads of data collection for those studies, loads, so I can spend the day just going through the hospital system trying to find information for the database. So I come back home and I'm drained again, tired. This is where I look for shortcuts when it comes to food. I have my nap, it's 6pm in the evening I'm not going to make a fancy meal, it needs to be something quick". (Participant 5)

Another relevant aspect related to work-life was the availability of unhealthy food in hospital settings. Most of the participants that worked in this context reported a normalised custom by patients, families and work colleagues to deliver high-content sugar and calorie food such as snacks, chocolates and cakes as a token of appreciation for their care and as a mean of 'fuel' their busy days. The constant availability of this type of food in staff rooms and kitchens increased the exposure to unhealthy eating; for example, participant 1 explained this situation:

"Then obviously all the free food we've been getting which has not been helping or the free chocolate–we got a massive order the other day of just chocolate upon chocolate upon chocolate. Sweets. Which is lovely and we're all incredibly grateful but it does make it a little bit harder". (Participant 1)

### 3.2 People interacting with digital media for positive practice change

The interviewees deepened their relationships with new technologies in their day-to-day life and uncovered different characteristics and ways in which they positively influenced. The participants identified the role of digital media in shaping healthy lifestyle knowledge, which could lead to positive changes in dietary and exercise practices. For example, participant 17, who used to have obesity as a young adult, highlighted a transformation due to his fitness journey. Constant physical training, the motivation to become a personal trainer and getting

helpful content on YouTube on physical exercise and different types of diets supported a positive, healthy lifestyle change. Also, participant 2 mentioned how the use of different technological platforms allowed her access to free, inspirational, reliable and scientific information, also with the thought that this information could be transferred and applicable to her circumstances. Consequently, she felt more confident in attempting to modify her cooking practices and perform physical exercises:

> "I was very aware of obesity due to my past, and I decided to become a personal trainer. And apart from that, I spend much time on YouTube, getting informed. You start looking at workouts, then you start looking at diets, then you start looking at supplements, and then you end up looking at what a lifestyle is. So maybe my point of view is very different from how the rest of the population, who may not invest their free time in knowing about these things. I like it; I enjoy it". (Participant 17)

> "I think one of the main things that started me changing the way I was thinking about things was I found TED talks. I didn't know what that was, my goodness just when you are washing up or you are in the kitchen cooking you've just got a TED talk playing in the background just on random things about what you'd like to learn about, giving you proper scientific evaluation of things and it gives you more confidence to think I am right. The computers and the TV have come in brilliant because the amount of things that you can access for free like indoor exercise, exercise for older people which is useful for me even though I'm not old with yoga and things like that it's been really good for me and it's given me a lot of chance". (Participant 2)

Some participants did not explain the positive influence of digital media exclusively but the importance of specific digital celebrities and influencers in diffusing relevant information and, above all, how that information was transmitted. For example, participant 5 explained a YouTuber's positive influence since this person was very relaxed in his approach to preparing new cooking recipes. This way of communication motivated her to try to incorporate a change in her cooking practices:

> "She pointed me in the direction of this guy on YouTube and I hate watching YouTube videos, there is too much choice, there is just too many people doing those things. I liked this bloke because he was very relaxed in his approach, he wouldn't preach me on the type of flour I had to have, he was like do you know it doesn't really matter. A lot of it was very relaxed so I thought actually watching him saying oh I can do that it's possible". (Participant 5)

### 3.3 The need to prioritise prevention in schools, the National Health Service and the food industry

The interviewees were also asked about their strategic priorities if they were the Prime Minister of the United Kingdom to reverse the harmful effects of the environment on their health practices. All of them were clear about the need to focus much more on a preventive approach to three particular environmental influences: schools, the NHS and the food industry. Thus, the prioritisation of education in children was identified as the most important action. For example, participant 15 highlighted a need to change children's knowledge at schools, specifically, by providing education on cooking healthy and increasing the awareness of the content of food in terms of calories and types of nutrients. Participant 1 extended this thinking and added the importance of involving working parents in cooking lessons, specifically, to develop

knowledge about cooking healthy with a limited budget. For this participant, the loss of cooking skills is a current nationwide issue:

> "Start with the kids, with children and teach them how to cook properly, teach them how to cook at school, bring back sort of very basic home economics, but not just teach them how to cook, but teach them what is in food. So the calorie content of food to understand, you know, you shouldn't be going over a thousand eight hundred calories a day if you weigh this amount of weight". (Participant 15)

> "I would invest the money into schools, into education in food into schools–not schools so much but I think cooking lessons. Yes, I think if you can have a good relationship with food when you are younger and you can learn how to experiment with food, I don't know if I want to say for like working families. I think food is fun and I think people need to know that there is fun from food, but I think people just don't know how to cook. So I think that they go for the easy option, they don't learn. Parents aren't available to be able to teach them how to cook. I learnt how to cook as I got older, but I never learnt with my parents whereas my friends' dad is a fantastic cook and she learnt how to cook from him. I think if that was something that you gave people those opportunities to learn and taught them how to do it really basic like on very limited money, I think if it's like a mandatory thing you do with your parents at school as you grow up, like the parents come in, the kids come in and they learn to cook". (Participant 1)

According to participants' opinions, despite obesity being a significant health issue, the NHS seems far from tackling it effectively. Thus, the second most important priority was investing more money in the NHS to create more primary prevention services and work towards a change in the obesity management plans. For example, participant 11, who had a leadership role in the NHS, explained that the current strategies focus more on developing treatments and interventions for those affected and experiencing long-term conditions related to obesity (e.g. cardiovascular diseases) instead of developing measures for preventing excess weight and conducting healthier lifestyles. Participant 14 also highlighted the importance of prevention and added the need to contract more multidisciplinary specialists to deal with health practices and other causes and the wider consequences (e.g. biological and psychological) related to weight loss.

> "Not much investment on preventative medicines or primary prevention. You'd probably have more money to help people and put ICDs (Implantable Cardioverter Defibrillator) and very expensive heart surgery but no investment for people to live healthily, give them support to lose their weight, no investment to give them healthier options or fitness training or have the ability to actually have a very good work/life balance". (Participant 11)

> "Above all, strengthen the health system in terms of prevention and forms and give real money in terms of weight reduction, more dietitians, more nutritionists. Prevention based on tests when children start to gain weight. The entire NHS's vital part has to be forced to implement all this, including mental health, endocrines, tests, and primary care because primary care is totally forgotten". (Participant 14)

Changing the food industry by penalising junk food companies and investing more money in British agriculture to reduce basic products' prices were highlighted as potential measurements to influence peoples' decisions and encourage them to choose healthy products. For example, participant 13 established comparisons and reflected on the positive impact of

corrective taxes on alcohol and tobacco, which could have similar effects on unhealthy food. Participant 6 insisted on the importance of investing more in farmer's markets and British agriculture to reduce the importing of food from other countries, which could reduce the price for the consumers.

> "I think stronger curtailing of, and you know, high taxation and all those sorts of things of that food industry. Like cigarette and smoking, the same has to be applied I think to fast food, so it's actually not spending money, but actually it is making harder for food companies to profit from poor quality food and high fat high sugar". (Participant 13)

> "I will try to get the companies subsidise the company who are for the key ingredients like your milk, like your bread, like your vegetables rather than constantly bringing, a lot of the vegetables don't come from the United Kingdom so invest in agriculture. We can grow our own potatoes 30 years ago to feed why can we not do so, continue to do so and produce rather than to constantly buy because it's cheaper to buy from abroad and bring in. But it's cheaper for the company not for the consumer. So bring things more locally, invest a lot more in agriculture". (Participant 6)

## 4. Discussion

This qualitative study covers a current gap that asked for research on lay perspectives to understand the wider notion and the influence of the obesogenic environment on adults with personal trajectories of living with obesity in the UK to adopt health practices related to eating and physical activity. The generated themes represent information about barriers, opportunities, and priorities to change within the environment.

The first theme describes the food environment as the most important barrier to conducting healthy practices and presents different ways people interact with it. For example, the increased availability and permanent exposure to unhealthy and cheap food in local stores, supermarkets and restaurants might encourage unhealthy food choices. Other studies showed a substantial rise in exposure to food outlets and foods for consumption away from home in the UK [53, 54]. Further research identified how retail food environments promote less healthy food by presenting a more significant reduction in price for a set cost than promotions on healthy drinks and food [55]. In this line, Coker et al. [56] showed that people who buy more of their food on promotion tend to purchase more High in Fat, Salt or Sugar (HFSS) products and are more likely to be people with excess weight. Something new was the identification of contradictions in the participants' narratives when justifying shopping and eating practices using a limited budget. Some people with obesity identified that eating healthier was more expensive than eating junk food using a limited budget, an idea rejected by people who had already lost weight.

The demands of working life were facilitators to increasing access to convenient food. Thus, some current work practices make it challenging to find time and energy to eat and cook healthier. The results might uncover the presence of employment inequalities, suggesting that people with specific jobs (e.g. nurses) might have less flexibility over working hours and can spend less time on self-care, mirroring the findings of other health conditions' research [57, 58]. Also, participants identified hospitals as snacking environments, considering them as negative environmental influences. This differs from the current public perspectives that consider the NHS part of a separate management system with no negative influence on individuals' health practices. Other research points out barriers to eating healthy in the workplace (without specification), such as eating more unplanned junk food in response to stress, co-worker influence and time constraints [59].

The second theme describes inter-relationships of people with new technologies (e.g. social media platforms such as YouTube, TED Talks, or documentaries on Netflix) and digital celebrities to improve their health practices related to eating and physical activity. While our study uncovered some instances where participants viewed new technologies as negative influences, we believe that emphasising their predominantly positive influence, which was stressed far more frequently by the interviewees, could offer a fresh perspective. This is particularly relevant because social media environments have traditionally been considered to have a negative role, for example, through constant exposure to junk food advertisements [60, 61], screen media exposure leading to a displacement in physical activity levels and an increase in energy intake from eating while sitting [62]. Also, the use of social network sites (e.g. YouTube or Instagram) for beauty and body norms can lead to binge eating disorders and excess weight as a consequence of negative individual feelings for not obtaining the weight goal [63] and compulsive workouts with significant abnormal eating [64–66] and potential loss of control [67]. Little research has identified the positive effects of social media platforms, social network sites and the influencers part of them. Some studies evaluated the effectiveness of community-cooking skills education programs delivered by television celebrity chefs (e.g. Jamie Oliver), which are being used to promote cooking confidence and skills as a vehicle for healthy eating with positive results [68, 69]. In addition, a recent scoping review studied the relationship between social media and physical activity and identified that one-third of the studies revealed positive effects regarding the promotion of physical activity and other health outcomes such as weight loss and blood pressure reduction [70]. Our results extend these positive influences by showing how technology and digital celebrities can induce positive changes in obesity-related health practices (e.g. through the availability of innovative and inspirational information and how information is transmitted). The last theme identifies preventive priorities in three particular environments to change the current obesity trend in the UK: schools, the NHS and the food industry. We reflect on these priorities later when discussing future implications by comparing them with previous policy interventions and potential future directions.

These themes and what has been done before to tackle obesogenic environments ask for an in-depth reflection on future implications. The information from the first theme (e.g. the contradictions about eating healthy and price) suggests that there might be gaps in people's capacity to interact with obesogenic environments designed to promote unhealthy choices. Providing further knowledge and psychological skills for self-control [71] to protect their vulnerability against the constant exposure to what the obesogenic environments offer (e.g. HFSS promotions in supermarkets or how food stores work to encourage impulsive decisions) seems pertinent. Thus, people could make better decisions within these environments and adjust their practices to avoid their negative impact (e.g. increasing knowledge of types of nutrients or planning and preparing healthy meals for the week). Part of that knowledge could be accessed through the infinite possibilities that new technologies, social media, and influencers offer. Social media and digital tools could promote and facilitate access to healthy lifestyles, especially in populations with health inequalities that suffer from geographical, financial or educational barriers. Knowledge and cognitive skills can be changed; however, it needs to be considered that these adjustments for practice change might not always be possible or would require extra effort from people due to inequitable social structures. Some examples could be employment inequalities (e.g. time to dedicate to improving health and well-being) [72, 73], a severe lack of household economic capabilities that allow only the choice of the necessary (people at high risk of food insecurity) [74, 75] or close access to more expensive and small food retailers rather than supermarkets (the latter offer more variety of food products and prices) [76, 77].

Bringing to light stories like those of the health workers who talked about the demands of working life and the hospital snacking environments could have implications. For example,

they could raise awareness of the constant exposure to HFSS food at workplaces and open the door to change the social norms of providing this type of food as a sign of gratitude by patients, family and co-workers. Also, these stories could make more visible the nature of specific employments (e.g. nurses and their long day and night shifts and stress) and how they might be linked with health inequalities (e.g. more risk of developing excess weight). Despite the potential and unmodifiable nature of the everyday stress and pace of working life, measurements could be put in place to protect employees' health and well-being. For example, providing protected time to conduct healthier lifestyles during workdays and implementing employee health screenings could help identify health risk factors and promote healthy living. These results could avoid unfair critiques towards the physical health of healthcare staff [78] that focus on individual responsibility and do not consider the role of environmental influences.

The identified priorities by the participants to change aspects of the current obesogenic environment should be considered to understand what measures currently exist and what people know about them, what measures are working or not, and propose future actions. First, providing education in school environments to children and family was identified as the main priority although it is acknowledged there are many facets of the school environment that are likely to be implicated (e.g. types of cooked food, vending machines or access to food outlets). There has been significant progress in British schools in terms of education on healthy eating in children and families based on a limited budget. For example, The National Curriculum in England implemented the subject 'Cooking and nutrition' in schools to teach students about food provenance and origins, cooking and food preparation, and applying healthy nutrition and eating [79]. However, it is necessary to allocate more time and resources to food education knowledge, which is still inconsistent across primary and secondary schools [80], something that can affect future practices [81].

Another priority asked for a change in how the NHS approaches obesity, with a need to focus more on prevention than treatment options and support people living with obesity by incorporating more multidisciplinary specialists. Lately, recent NHS reports have developed strategies to improve obesity management [82–84]. The plan is that the NHS will focus more on prevention, including obesity-related ill health and improving services for suppl behaviour change via information provision, service design and clinical interventions (e.g. targeted weight management services or social prescribing). Ensuring that people with obesity and front-line professionals are aware of all the new and future services available will be essential. We suggest that providing continuous and up-to-date education for patients and even professionals about the role of positive and negative environments could be considered since it seems that most weight management interventions focus exclusively on the individual and ignore those complex environments that are constantly changing and adapting to business needs.

The last priority asked for the control of the food industry and food prices. In this line, the British government has designed and implemented some strategies to tackle obesity [85–89], which seem insufficient. Thus, some authors state that there has not been enough progress since the strength of industry opposition and government hesitation to implement interventionist policies to force restraints on the free market and influence individual choice presents a significant barrier [90–93]. Other authors identified the inability of policymakers to regulate processes and environments relevant to chronic illness management [94]. Therefore, we suggest that working on how the individuals relate to the environments and giving them the tools to make healthier choices seems more pertinent than waiting for a significant and effective response from the government and the industry.

In terms of future research and considering that the environment is constantly changing, qualitative research needs to be a continuous priority to uncover the different environmental

influences that operate across multiple levels of society [10]. Qualitative research applied to understand interactions between individuals and environments should prioritise people living in disadvantaged circumstances (e.g. low income, less education, limited access to space or more exposure to the sale of unhealthy foods) since these factors impact whether people can eat healthily or be active and increase the risk of developing excess weight [95]. In this line, understanding and compiling lay perspectives and other members of the public (e.g. health-care professionals) perspectives about the impact of obesogenic environments could play a crucial role in developing equitable interventions and policies where health inequalities are addressed. The role of social media and social media influencers seems to be an area that needs further exploration. For example, researching the processes of interaction between people looking for health changes in social media, how and why people engage or do not with them, how information is diffused, and the identification of role models through opinion leaders theory might be relevant. Finally, the findings of this study emphasise the need to investigate the food environment and eating practices within the workplace, a factor that appears to be under-represented in current research on obesity in the workplace setting. Much of the existing litera-ture focuses on sedentary work, stress, shift work, and extended working hours as primary contributors to excess weight gain among employees [96, 97]. However, it is crucial to recog-nise that the workplace is an environment where individuals engage in various social interac-tions, many of which revolve around food and eating (e.g. communal meals, celebratory events and the type of food available, informal gatherings or pressure from others to try certain foods). By exploring the social and cultural norms that shape eating practices in the workplace, researchers could develop a deeper understanding of how these norms impact employees' food choices and overall dietary patterns [98].

## 4.1 Strengths and limitations

One of the strengths of this research is the use of personal accounts to uncover current pro-cesses of interaction between individuals and multiple environments and identify overlooked environmental factors that could influence the adoption of different lifestyles in adults with obesity, an aspect that quantitative studies cannot do.

Another strength is including healthcare professionals with a history of obesity as lay involvement. Having worked as clinicians recently, the researchers realised that obesity was normalised by many healthcare professionals and did not see it as a health risk. They can be considered lay people in so far as obesity, food, and eating practices are not at the centre stage on the nursing curriculum, and a recent epidemiological study shows the high obesity rates among healthcare workers [99]. Their views were no different in terms of technical knowledge from the accounts given by the non-health professionals. Thus, their experiences in their everyday working environment (hospital as the micro-environment and the NHS healthy sys-tem as the macro-environment) are innovative and relevant to the research aim.

On the other hand, this research and its development context show challenges and limita-tions. The coronavirus pandemic, the lockdowns, and the fact that this topic encompasses sig-nificant social stigma challenged the recruitment process and the possibility of attracting more interested potential participants. Forty-four local and national weight management groups and obesity and long-term conditions charities were contacted with no success. The majority did not reply to the requests, a few just declined the potential participation without providing reasons, and others stated that the lockdown period was not the moment to conduct this type of research. Also, we could not obtain a varied sample of participants (e.g. different socio-demographic characteristics), so we did not explore the relationship between the individuals' attributes and their attached health opinions. Therefore, our results must be considered

cautiously and not establish generalisations (results are not transferable) since they might not apply to people with other socio-demographic characteristics.

## 5. Conclusions

This qualitative study explores how adults with a history of obesity interact with multiple environments, which could shape the adoption of different individual eating and physical activity practices. The food environment seems to be the major problem due to the disproportionate amount and low price of unhealthy products and the demands and inequalities of working life as a facilitator to access them. Specific workplaces, such as hospital settings, can become snacking environments due to established social norms. On the other hand, social media and digital celebrities and how they interact with individuals and vice-versa could promote positive health changes. Schools, the NHS and food companies were targeted as negative environmental influences and the main settings to prioritise preventive measures against obesity. The results uncover new environmental factors and relational aspects between adults and the environment based on everyday events that influence the adoption of health practices. These perceptions from individuals living with obesity could inform the design of more adequate and effective interventions that consider, more than in the past, the interactions between the environment and individuals. Finally, this research could raise awareness amongst lay populations about the socio-ecological nature of obesity, reducing blaming and stigma from society and creating a more conducive context for political and societal change.

## Supporting information

**S1 Appendix.**
(DOCX)

**S2 Appendix.**
(DOCX)

## Acknowledgments

We want to thank all the participants for accepting being interviewed on such a sensitive topic and during challenging times (COVID-19 pandemic). Finally, we would like to thank Jackie Barney for her help with the transcripts.

## Author Contributions

**Conceptualization:** Nestor Serrano-Fuentes, Anne Rogers, Mari Carmen Portillo.

**Data curation:** Nestor Serrano-Fuentes.

**Formal analysis:** Nestor Serrano-Fuentes.

**Investigation:** Nestor Serrano-Fuentes.

**Methodology:** Nestor Serrano-Fuentes.

**Project administration:** Nestor Serrano-Fuentes.

**Supervision:** Anne Rogers, Mari Carmen Portillo.

**Validation:** Anne Rogers, Mari Carmen Portillo.

**Writing – original draft:** Nestor Serrano-Fuentes.

**Writing – review & editing:** Nestor Serrano-Fuentes, Anne Rogers, Mari Carmen Portillo.

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
