## [Decision Letter · Decision Letter 0]

6 Feb 2024

PONE-D-23-31021Beyond individual responsibility: exploring lay understandings of the contribution of environments on personal trajectories of obesityPLOS ONE

Dear Dr. Serrano-Fuentes,

Thank you for submitting your manuscript to PLOS ONE. After careful consideration, we feel that it has merit but does not fully meet PLOS ONE’s publication criteria as it currently stands. Therefore, we invite you to submit a revised version of the manuscript that addresses the points raised during the review process.

We look forward to receiving your revised manuscript.

Kind regards,

Fernanda Penido Matozinhos, Ph.D

Academic Editor

PLOS ONE

Journal Requirements:

Additional Editor Comments:

Dear Author,

After careful consideration, I feel the manuscript explores a very important topic. The next modifications in the text are going to make the manuscript come to a better result.

Kind regards.

Reviewers' comments:

Reviewer's Responses to Questions

**Comments to the Author**

1. Is the manuscript technically sound, and do the data support the conclusions?

Reviewer #1: Yes

Reviewer #2: Yes

2. Has the statistical analysis been performed appropriately and rigorously? 

Reviewer #1: Yes

Reviewer #2: Yes

3. Have the authors made all data underlying the findings in their manuscript fully available?

Reviewer #1: No

Reviewer #2: Yes

4. Is the manuscript presented in an intelligible fashion and written in standard English?

Reviewer #1: Yes

Reviewer #2: Yes

5. Review Comments to the Author

Reviewer #1: This article explores the impact of environment on obesity and shows rigorous methods towards analysis and interpretation of the data. I really enjoyed reading this article and have some very minor revisions/comments/considerations for the author:

1. Data availability - the authors state the data is available but there is no indication of where or how to access the data. Please add in a DOI or relevant pathway to a repository

2. Some of the in-line references are in a different format from the remainder of the manuscript. see lines 47, 73, 496/497

3. rephrasing of line 96 (parents of the environment or of the kids?)

4. Line 122 - this is a little vague. How were the researchers assumptions included?

5. Line 129 - sampling strategy unclear. It states both purposeful and snowball which sounds like convenience?

6. How were advocacy individuals identified? where were they from? healthcare, charities?

7. line 144 - were participants 1 and 2 involved in both the development/review of the schedule and involved in the interview? if so, could this have affected their answers?

8. Table 1 - 6 participants have a BMI below 30. Was there a time limit (i.e. within the past X years) that they had to have lived with obesity? I think this is an interesting consideration given how much our environments have changed recently (e.g. covid, becoming more digitised)? Were these participants still actively managing their weight?

9. line 431 - there is existing literature on the role of digital tech in weight management. Important to distinguish between tech and social media. For example: doi: https://doi.org/10.1136/bmj.p1882 There are also UK-based weight management social platforms which have published results. You are right that more research is needed, but there is evidence out there.

10. line 466 - any studies to support this point? types of employment and health inequalities

lines 521/527 - reference?

General

1. Lack of reflexivity in the manuscript. How could have the researchers role and experience have influenced the participants?

2. Discussion - specifically, I think discussing the employment side further could be interesting. There is a slight bias in this towards healthcare (which is really important) but do you think there are other types of employment that could be explored or could have an effect on obesity. this may be of interest: DOI: 10.1007/s40273-014-0243-x

Reviewer #2: This manuscript is very interesting, thoughtful, and well-written. I agree with the authors regarding the importance of looking into the environment to handle obesity comprehensively.

Two things might need to be explained further. First, the need to explore the possible different experiences/perspectives of people who have overcome obesity and people who are currently obese. Second, the social media analysis didn't mention any negative issues. Please clarify whether there were no such issues or the authors didn't include them in the analysis and discussion.

6. PLOS authors have the option to publish the peer review history of their article (what does this mean?). If published, this will include your full peer review and any attached files.

Reviewer #1: No

Reviewer #2: **Yes: **Nani C Sudarsono

---

## [Author Response · Author response to Decision Letter 0]

15 Mar 2024

REVIEWER 1

1. Data availability - the authors state the data is available but there is no indication of where or how to access the data. Please add in a DOI or relevant pathway to a repository.

Answer: 1. All the data is available in the manuscript and supporting information (Appendix). That is specified in the manuscript. There is no more data to share (e.g., transcripts of the interviews); therefore, no data has been uploaded to, for example, repositories. 

2. Some of the in-line references are in a different format from the remainder of the manuscript. see lines 47, 73, 496/497

Answer 2: The in-text citations have been adapted to the Vancouver style.

3. rephrasing of line 96 (parents of the environment or of the kids?)

Answer 3: Line 96 has been clarified

4. Line 122 - this is a little vague. How were the researchers assumptions included?

Answer 4: We appreciate the reviewer's insightful comment regarding the statement, "The researchers' assumptions were not bracketed or set aside but were embedded and essential to the interpretive process." We would like to provide further clarification on this point, taking into account that reflexive thematic analysis involves coding without attempting to fit the data into the researcher's preconceived analytical notions. We think that these two statements are not mutually exclusive. While our assumptions were not considered in the initial coding process, they played a role in the development of the final themes and the narrative we aimed to present as part of the reflexive and inductive process. Let’s explain this. 

For instance, based on our familiarity with the literature, we were aware that social media is often portrayed as a negative influence, such as through the constant exposure to unhealthy food advertisements. We had assumed that participants would primarily highlight these negative aspects. However, we also recognised that there is limited research exploring the positive influence of social media. Surprisingly, during the analysis, we discovered that participants placed a greater emphasis on the positive impact of social media.

When identifying the various environmental opportunities, our goal was to present a novel perspective in the results section. Consequently, the development of the theme "new technologies as inducers of health practices" and its accompanying narrative was a deliberate choice made during the analysis process. This decision was influenced not only by the data itself but also by our familiarity with the existing literature on this topic, which challenged our initial assumptions.

We have explained this in the S1 Appendix (phase 5)

5. Line 129 - sampling strategy unclear. It states both purposeful and snowball which sounds like convenience?

Answer 5: We think that the sample is purposeful and not convenient. Purposive sampling refers to intentionally selecting participants based on their knowledge, characteristics, and experiences, among others. Convenience sampling involves recruiting individuals primarily because they are willing, available, or easy to access or contact on a practical level. Convenience sample could incorporate participants who have not had the experience of living with obesity. We have rephrased that section to clarify the importance of considering individuals who have experienced living with obesity. 

6. How were advocacy individuals identified? where were they from? healthcare, charities?

Answer 6: Information added in lines 139-141.

7. line 144 - were participants 1 and 2 involved in both the development/review of the schedule and involved in the interview? if so, could this have affected their answers?

Answer 7: There was not a proper Patient and Public Involvement (PPI) before the interviews, for example, creating a group of a few citizens who could provide feedback on the research topic, aim, research adverts or interview questions since this was a PhD project with lack of financial resources to conduct this step. The interview questions were pilot-tested with the first two participants, who provided feedback on the questions after they conducted their interviews. We have clarified those lines in the text (145-148).

8. Table 1 - 6 participants have a BMI below 30. Was there a time limit (i.e. within the past X years) that they had to have lived with obesity? I think this is an interesting consideration given how much our environments have changed recently (e.g. covid, becoming more digitised)? Were these participants still actively managing their weight?

Answer 8: There wasn’t a time limit. The idea was to recruit participants who had obesity at some point in their lives.

It’s very relevant the “time” dimension that you are recommending. We addressed that in a different article we had already published that considered the same sample and other questions from the interview schedule (1).

1. Serrano-Fuentes N, Rogers A, Portillo MC. The influence of social relationships and activities on the health of adults with obesity: A qualitative study. Health Expect Int J Public Particip Health Care Health Policy. 2022;25(4):1892–903. 

For that article, we studied the influence of social relationships on health practices over time and how that could have an impact in the present. The interview questions for this current article do not address that temporal dimension (also, there were no relevant answers from the participants in this sense), but more the current obesogenic environment. 

Also, we did not consider or ask if these participants (who are currently below a BMI of 30) are currently managing their weight. Some participants talked about it during the interview when discussing their personal experiences, and some insights can be seen in the already published article we specified above. That information is not relevant to this particular article. 

We hope this makes sense to the reviewer. 

9. line 431 - there is existing literature on the role of digital tech in weight management. Important to distinguish between tech and social media. For example: doi: https://doi.org/10.1136/bmj.p1882 There are also UK-based weight management social platforms which have published results. You are right that more research is needed, but there is evidence out there.

Answer 9: Thank you very much. Indeed, we wanted to refer to social media platforms, social network sites and the celebrities part of them. At no time did we want to reference specific weight management new technologies, such as apps or online courses. We are aware that there are lots of resources like that, for example, in the NHS, as part of the Better Health Programme, and that there is already research evaluating those services. We have clarified those lines to avoid confusion. 

10. line 466 - any studies to support this point? types of employment and health inequalities

Answer 10: We have added some references in the last part of the paragraph to support the narrative. Lines 478-481

11. lines 521/527 - reference?

Answer 11: References have been added in lines 532-538 concerning the importance of qualitative research.

General: 1. Lack of reflexivity in the manuscript. How could have the researchers role and experience have influenced the participants?

Answer: We have added a paragraph (the last paragraph of the data collection section) about how we tried to avoid biased answers based on our role. We had also created the S1 Appendix precisely to explain aspects of reflexivity during the analysis. This can be seen in the detailed analysis phases (e.g., how codes and themes were developed and changed), as well as detailed aspects of the quality and rigour of data at the end of the appendix. As seen in one of the previous comments, we added in phase 5 an explanation of how the researchers' assumptions were not bracketed or set aside but were embedded and essential to the interpretive process. We hope that all this information (also the S2 appendix) is enough to show detailed aspects of reflexivity of this qualitative study. 

2. Discussion - specifically, I think discussing the employment side further could be interesting. There is a slight bias in this towards healthcare (which is really important) but do you think there are other types of employment that could be explored or could have an effect on obesity. this may be of interest: DOI: 10.1007/s40273-014-0243-x

Answer: 2. Thank you for the suggestion. There is already lots of research showing the relationship between the workplace/jobs and obesity (sedentary work, stress, shift work and long working hours), but very few regarding the NHS. An aspect that we think needs to be explored further is the food environment through social practices, social and cultural norms of eating, etc., which, in some way, we have addressed regarding the NHS (e.g. as a snacking environment and family members providing unhealthy food as a way of gratitude). We have added information in this regard in the future research paragraph (discussion section), lines 546-556. 

REVIEWER 2.

This manuscript is very interesting, thoughtful, and well-written. I agree with the authors regarding the importance of looking into the environment to handle obesity comprehensively.

Two things might need to be explained further. First, the need to explore the possible different experiences/perspectives of people who have overcome obesity and people who are currently obese. Second, the social media analysis didn't mention any negative issues. Please clarify whether there were no such issues or the authors didn't include them in the analysis and discussion.

Answers: Thank you very much for the feedback. 

In terms of exploring the differences between people with obesity and people who used to have obesity, we must say that we looked into that. However, we only found one difference, which was highlighted in another article we wrote studying the same sample and addressing other interview questions, which explored the influence of social relationships (as part of the obesogenic environment). The difference was that most of the people who managed to lose weight (therefore, were successful in their weight loss journey) had more varied networks (relationships) with the inclusion of community interactions (e.g. gym friends, weight management group’s friends, etc.). This can be seen here:

Serrano-Fuentes N, Rogers A, Portillo MC. The influence of social relationships and activities on the health of adults with obesity: A qualitative study. Health Expect Int J Public Particip Health Care Health Policy. 2022;25(4):1892–903.

In the current article, which explores other aspects of the food environment, we did not identify any significant differences in the responses provided by the two groups. This lack of observable differences may be attributed to the fact that our questions in this study did not delve into the temporal aspects and how these environments evolve (we did this on purpose). Perhaps, investigating the time-related changes in these environments could have potentially revealed more distinctions between the two groups (but this is just a vague idea). 

Concerning social media influence, we also found some examples indicating their negative influence (mentioned in 7 articles, as can be seen in the S1 Appendix), but there were fewer than those indicating a positive influence (mentioned in 12 articles). We have added some information in this regard in the discussion, starting in line 435. “While our study uncovered some instances where participants viewed new technologies as negative influences, we believe that emphasising their predominantly positive influence, which was stressed far more frequently by the interviewees, could offer a fresh perspective. This is particularly relevant because social media environments have traditionally been considered to have a negative role, for example, through constant exposure to junk food advertisements (60,61)…”

The way we wrote our results specifying only their positive influence was on purpose, but we hope this further explanation in the discussion provides more context and information for the readers. 

Thank you.

---

## [Decision Letter · Decision Letter 1]

15 Apr 2024

Beyond individual responsibility: exploring lay understandings of the contribution of environments on personal trajectories of obesity

PONE-D-23-31021R1

Dear Dr. Nestor Serrano-Fuentes,

We’re pleased to inform you that your manuscript has been judged scientifically suitable for publication and will be formally accepted for publication once it meets all outstanding technical requirements.

Kind regards,

Fernanda Penido Matozinhos, Ph.D

Academic Editor

PLOS ONE

Additional Editor Comments (optional):

Dear Author,

After careful consideration, I feel the manuscript explores a very important topic. The questions were responded and modifications in the text made the manuscript come to a satisfying result. We recommend its publication.

Kind regards,

Reviewers' comments:

Reviewer's Responses to Questions

**Comments to the Author**

1. If the authors have adequately addressed your comments raised in a previous round of review and you feel that this manuscript is now acceptable for publication, you may indicate that here to bypass the “Comments to the Author” section, enter your conflict of interest statement in the “Confidential to Editor” section, and submit your "Accept" recommendation.

Reviewer #1: All comments have been addressed

Reviewer #2: All comments have been addressed

2. Is the manuscript technically sound, and do the data support the conclusions?

Reviewer #1: Yes

Reviewer #2: Yes

3. Has the statistical analysis been performed appropriately and rigorously? 

Reviewer #1: Yes

Reviewer #2: Yes

4. Have the authors made all data underlying the findings in their manuscript fully available?

Reviewer #1: Yes

Reviewer #2: Yes

5. Is the manuscript presented in an intelligible fashion and written in standard English?

Reviewer #1: Yes

Reviewer #2: Yes

6. Review Comments to the Author

Reviewer #1: (No Response)

Reviewer #2: The article has been revised as per the request, and the authors have incorporated all the feedback. I hope this article will help deepen the understanding of obesity, particularly regarding environmental issues. The study's emphasis on social media is refreshing and makes it current.

7. PLOS authors have the option to publish the peer review history of their article (what does this mean?). If published, this will include your full peer review and any attached files.

Reviewer #1: No

Reviewer #2: **Yes: **Nani C Sudarsono

---

## [Editor Report · Acceptance letter]

26 Apr 2024

PONE-D-23-31021R1 

PLOS ONE

Dear Dr. Serrano-Fuentes, 

I'm pleased to inform you that your manuscript has been deemed suitable for publication in PLOS ONE. Congratulations! Your manuscript is now being handed over to our production team.

Kind regards, 

on behalf of

Dr. Fernanda Penido Matozinhos 

Academic Editor

PLOS ONE